# MusiXQA: Advancing Visual Music Understanding in Multimodal LLMs

## Abstract

Multimodal Large Language Models (MLLMs) have achieved remarkable visual reasoning abilities in natural images, text-rich documents, and graphic designs. However, their ability to interpret music sheets remains underexplored. To bridge this gap, we introduce MusiXQA, the first comprehensive dataset for evaluating and advancing MLLMs in music sheet understanding. MusiXQA features high-quality synthetic music sheets generated via MusiXTeX, with structured annotations covering note pitch and duration, chords, clefs, key/time signatures, and text, enabling diverse visual QA tasks. Through extensive evaluations, we reveal significant limitations of current state-of-the-art MLLMs in this domain. Beyond benchmarking, we developed Phi-3-MusiX, an MLLM fine-tuned on our dataset, achieving significant performance gains over GPT-based methods. The proposed dataset and model establish a foundation for future advances in MLLMs for music sheet understanding. Code, data, and model will be released upon acceptance.

## 1 Introduction

Multimodal Large Language Models (MLLMs) are becoming general-purpose reading assistants for visual content, capable of interpreting natural images and text-rich documents (Adobe Inc., 2024). However, existing models still struggle with visual question answering tasks involving music sheets, performing at near-random levels (Yue et al., 2024), indicating that this modality remains underexplored. Enabling MLLMs to read and reason over music sheets would be a valuable extension of their capabilities, as sheet music plays a central role in how music is taught, analyzed, and communicated. Although music notation is a symbolic and lossy approximation of sound, music sheets remain the only widely accepted visual system for the written transmission of music. It serves as a crucial bridge between audio, visual, and textual representations, making it an essential modality for large language model-based AI systems to effectively interpret and understand music. Unlike tasks such as reading visual text or recognizing objects in images, where answers are often apparent to the human eye, reading music notation requires interpreting dense symbolic structures (Meixner, 2015). Even for humans, achieving proficiency in reading music typically requires years of dedicated training (Behmer Jr & Jantzen, 2011). As a result, music sheet understanding poses a uniquely complex challenge, where AI assistance is not only beneficial to human but arguably more necessary than in many other visual understanding tasks.

The traditional approach to this task is Optical Music Recognition (OMR)(Shatri & Fazekas, 2020; Calvo-Zaragoza et al., 2020), a field with a long research history(Rebelo et al., 2012; Bainbridge & Bell, 2001; Fujinaga, 1988). Many existing OMR systems still rely on relatively small neural networks and multi-stage pipelines tailored to specific sub-tasks. In contrast, recent advances in MLLMs have shown strong performance across a range of visual tasks (Nguyen et al., 2024; Kuang et al., 2024; Chen et al., 2024d; Wang et al., 2024; Chen et al., 2024a), demonstrating that modern architectures can perform end-to-end reasoning over structured visual inputs directly in natural language format, without relying on traditional detection modules such as Optical Character Recognition (OCR) tools (Du et al., 2020; Singh et al., 2021). These developments suggest that MLLMs offer a promising alternative to pipeline-based approaches in domains like music sheet understanding, where such capabilities remain underexplored.

To bridge this gap, we introduce MusiXQA, a large-scale benchmark dataset designed to evaluate and enhance MLLMs for music sheet understanding. Figure 1 provides an overview of our data

Figure 1: Data generation and model workflow in Musi𝕏QA. Music metadata is sampled and rendered into sheet images via MusiXTEX, with QA pairs generated from templates. The resulting data is used to train and evaluate MLLMs on visual music understanding tasks.

generation and model inference workflow. Musi𝕏QA contains 9,600 high-quality synthetic music sheets rendered via MusiXTEX, paired with over 130,000 visual question-answer (QA) pairs spanning OCR, layout understanding, optical music recognition, and chord estimation. Unlike prior datasets, Musi𝕏QA offers diverse and balanced coverage of musical concepts, enabling fine-grained evaluation across multiple tasks. In addition, we develop Phi-3-MusiX, a fine-tuned version of Phi-3-Vision, adapted using parameter-efficient LoRA training on the Musi𝕏QA dataset. Experimental results show that existing MLLMs, including GPT-4o and Paligemma2, struggle with music sheets reading, while Phi-3-MusiX achieves up to eight times performance gains in GPT evaluation accuracy on the OMR-based task. These results highlight the importance of domain-specific supervision and the value of Musi𝕏QA as a resource to enable visual music understanding in MLLMs. Our contributions are as follows:

- We propose Musi𝕏QA, the **first** large-scale, diverse, and balanced synthetic dataset for visual question answering on music sheets.

- We develop Phi-3-MusiX, the **first** MLLM fine-tuned for music sheet understanding, which significantly outperforms the best baseline on our benchmark, demonstrating the effectiveness of our dataset in advancing symbolic music reasoning in MLLMs.

- We introduce a MusiXTEX based framework for scalable generation of music sheets, supporting both random/controlled data sampling and conversion from real-world MIDI data.

- We propose `kern+`, a compact symbolic representation designed for efficient modeling of pitch and duration, and analyze how output format influences training dynamics and model performance.

## 2 RELATED WORK

### 2.1 MUSIC SHEET BENCHMARKS AND DATASETS

The development of OMR and music sheet understanding has been supported by various datasets, each addressing different aspects of the task. CVC-MUSCIMA (Fornés et al., 2012) and MUS-CIMA++(Shatri & Fazekas, 2024) focus on handwritten music scores, tackling challenges in staff line removal and symbol segmentation. DeepScores(Tuggener et al., 2018) shifts toward typeset music, enabling fine-grained OMR tasks. For symbolic recognition, PrIMuS (Calvo-Zaragoza & Rizo, 2018) introduces monophonic scores with semantic and agnostic representations, while Camera-PrIMuS simulates real-world distortions in sheet music capture. DoReMi (Shatri & Fazekas, 2021) further explores single-line typeset music with note variations but omits key and time signature annotations. More recent efforts aim at structured evaluation. COMREF (Torras et al., 2024) provides over 400k measure-level typeset scores, using Music Tree Notation (MTN) for a standardized output representation. OLiMPiC (Mayer et al., 2024) offers scanned and synthetic system-level sheets, while MMMU (Yue et al., 2024) evaluates MLLMs via QA over system-level music scores, revealing performance close to random guessing. Despite these advances, existing datasets often exhibit distributional bias due to their reliance on real-world sources. In contrast, our dataset is synthetically generated with controlled diversity across musical attributes such as key, note density, and measure length. It provides a large-scale, full-page QA benchmark tailored for MLLMs, supporting comprehensive evaluation across multiple tasks.

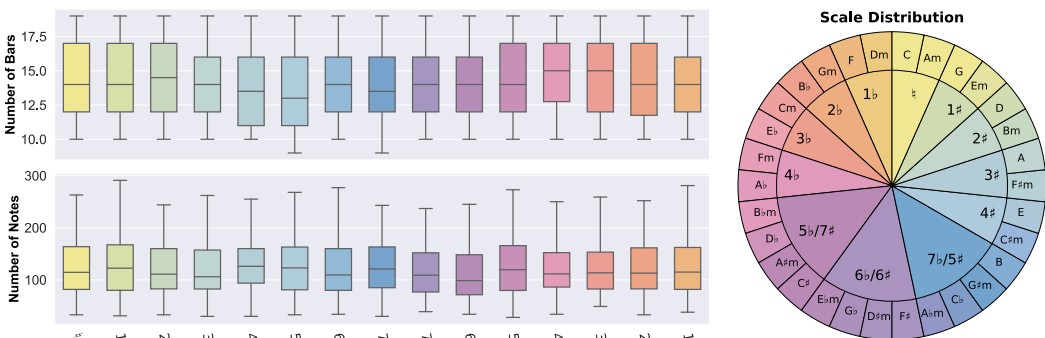

Figure 2: Data distribution of the Musi𝕏QA dataset. **Left**: Boxplot showing the distribution of the number of notes and bars per image. Each box represents the inter-quartile range (IQR), covering the middle 50% of the data. The horizontal line inside each box indicates the median document length, while the whiskers extend to the minimum and maximum values within 1.5 times the IQR. **Right**: Distribution of scales in the dataset. The inner circle groups enharmonic equivalent and relative scales according to the circle of fifths (Clough & Myerson, 1986). These scales share the same seven pitches but differ in accidentals (e.g., C♯ vs. D♭) or scale type (e.g., C major vs. A minor). The outer circle represents root of scales, with minor scales labeled using a lowercase 'm'.

## 2.2 OPTICAL MUSIC RECOGNITION

OMR converts sheet music images into digital formats such as MIDI or MusicXML, aiming to capture the complexity and variability of musical notation. Traditional OMR systems often rely on modular pipelines that separately handle staff line detection, symbol classification, and structural analysis. For example, Oemer (Yoyo et al., 2023) employs two UNet models alongside SVM classifiers to isolate staff lines and classify symbols, while Simonetta et al. (Simonetta et al., 2024) use neural network classifiers to identify musical elements in digitized manuscripts. Although effective for narrowly defined tasks, these systems typically require extensive preprocessing, handcrafted features, or manual annotations, which limits their scalability and adaptability. To address these limitations, recent deep learning methods aim to streamline OMR by integrating the entire workflow into a single model. These end-to-end approaches often combine CNN-based feature extraction with Transformer-driven sequence modeling. Eelco et al. (van der Wel & Ullrich, 2017) proposed a CNN-based sequence-to-sequence model that processes full sheet music phrases with data augmentation. Zeus (Mayer et al., 2024) introduced a direct transcription method that uses a linearized MusicXML format, compressing visual notation into concise state-change tokens. TrOMR (Li et al., 2023) employs a Transformer-based architecture to improve polyphonic OMR performance in real-world scenarios. SMT (Ríos-Vila et al., 2024) further advances this line of work by combining a CNN encoder with a Transformer decoder to transcribe complex polyphonic scores, and SMT++ (Rıos-Vila et al., 2024) extends this capability to full-page pianoform scores without requiring separate layout analysis. Despite these advancements, most existing OMR systems remain focused on transcription accuracy and are tightly coupled with specialized architectures. As a result, they are limited in their ability to generalize to broader symbolic music understanding tasks or scale to more flexible, unified AI systems.

## 2.3 MULTIMODAL LARGE LANGUAGE MODELS

Most existing MLLMs are not trained for music sheet understanding. Text-only models like ChatMusician(Yuan et al., 2024) and tool-based agents like MusicAgent(Yu et al., 2023) lack visual input capabilities and cannot process sheet music. Paligemma2(Steiner et al., 2024) includes music scores in its training data, but lacks comprehensive evaluation and suffers from limited image resolution. Large models like GPT-4o (Hurst et al., 2024) and DeepSeek (Liu et al., 2024a) reject music sheet reading requests in their online interfaces. Currently, no MLLMs have been explicitly trained or systematically evaluated for music sheet understanding.

# 3 MUSIⅩQA DATASET

To create a large-scale and accurately annotated dataset for visual question answering over music sheets, we generate synthetic data using MusiXTEX(Taupin et al., 1993), a LaTeX-based typesetting system designed for rendering music notation. This approach ensures precise annotations while enabling scalable data generation. Each music sheet is first compiled into a PDF, and then converted into high-resolution images, serving as the final data points for recognition tasks. Since our focus is on factual question-answering rather than musical composition, the generated notes do not need to be musically coherent or aesthetically refined, but must remain structurally valid and interpretable. To achieve this, we employ a heuristic, theory-guided approach to generate random yet well-formed music notation, ensuring correct note placement, structured layouts, with diverse rhythmic patterns and key signatures. By leveraging MusiXTEX and a controlled data sampling strategy, we construct a diverse dataset that enables models to answer structured questions about music notation, advancing the capabilities of music sheet understanding in modern multimodal large language models.

## 3.1 DATASET STATISTICS

We constructed a dataset of 96k unique music sheets, evenly distributed across 30 scales, covering a pitch range from A♭1 to F♯6. The dataset comprises 1.3 million bars / measures and 11.7 million notes with pitch and duration annotation, as illustrated in Figure 2. To support diverse evaluation tasks, we generated 670k OMR-based QA pairs, 337k OCR-based QA pairs, 288k layout understanding QA pairs, and 47k chord estimation QA pairs, enabling comprehensive AI-driven music analysis.

## 3.2 MUSIC SHEET CONFIGURATION

To generate structured yet diverse music sheets, we randomly sample music sheet configurations for key elements to produce the corresponding LaTeX source code. Figure 3 illustrates typical components, including title, composer, clefs, key and time signatures, tempo, chord labels, and initial measures in both treble and bass clefs.

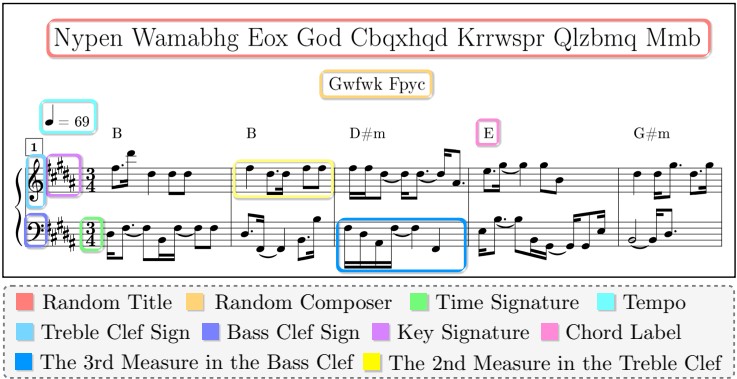

Figure 3: Example of key elements in a music sheet.

**Text Metadata** We randomize the title and author name to ensure privacy, copyright compliance, and dataset diversity. These elements, while not the focus of recognition, provide valuable OCR-related information. Titles consist of 1 to 10 words, and author names contain 1 to 3 words, with each word being 3 to 8 characters long, ensuring structural variability.

**Clef Settings** A clef defines the pitch range of the notes within the five-line system. The treble clef is commonly used for higher-pitched instruments, while the bass clef is used for lower-pitched ones. We generate music sheets in three configurations: (1) treble clef only, (2) bass clef only, and (3) both clefs, which are typically used for piano music.

**Tempo and Time Signature** Tempo and time signature define different aspects of musical rhythm. Tempo, measured in beats per minute (BPM), is randomly selected from 50 to 140 BPM and placed at the top of the first bar to indicate speed. The time signature, which determines beats per measure,

is randomly chosen from $[2, 3, 4]$, with a quarter note as one beat. We exclude 6/8 to simplify the encoding in MusiXTEX.

**Scale and Key Signature**  Each music sheet is assigned a scale name (e.g., C major), which specifies a set of notes based on a root note and a predefined pattern of intervals. The scale is reflected by its key signature, which is denoted by sharps (♯) or flats (♭) placed next to the clef signs, indicating which notes are consistently altered throughout the piece. To distinguish between a major scale and its relative minor scale (C major & A minor), we set the first bar using the tonic chord of the key, ensuring a clear tonal center. While Western music uses the twelve-tone equal temperament system (Mathieu, 1997) that divides an octave into 12 notes, our dataset includes enharmonic equivalent scales, which are scales that sound identical but are different in writing (e.g., C♯ major vs. D♭ major). We exclude keys requiring double sharps or double flats as they are rarely used in practice. This results in 15 distinct key signatures, providing broad symbolic diversity while maintaining real-world relevance. The pie chart in Figure 2 illustrates the distribution of scales. Table A.1 shows note compositions of major and minor scales.

### 3.3 CHORD-BASED MUSIC GENERATION

We generate music notes one bar at a time for 10 to 20 bars. For each bar, we first randomly select the number of notes, $n$, where $n$ is uniformly sampled between 1 and three times the number of beats in the bar. We then independently sample their durations and pitches, ensuring structural validity while maintaining notation diversity. The sampled notes are subsequently encoded as LaTeX code using MusiXTEX notation, enabling the automated generation of high-quality sheet music.

**Duration Sampling and Splitting**  For note durations, we divide each bar into bins of 16th notes and randomly group these bins into $n$ segments to determine note durations. This ensures that the total duration of all notes precisely sums to the required beats in the bar. After duration sampling, we further process the notes to ensure that they are correctly represented in standard music notation. Some durations cannot be notated as a single note; for example, three 16th notes must be written as a dotted 8th note rather than a standalone value. To handle such cases, we prioritize dotted notes and double-dotted notes (extending by 50% and 75%) whenever possible. If a duration still cannot be fully represented, we apply note splitting, dividing it into two tied notes as a last resort. To introduce notation variety, we apply two note grouping strategies with equal probability. In one approach, we use beat-based grouping, where notes within the same beat are beamed for improved rhythmic clarity. Additionally, if a note spans two beats, we split it into tied notes. In the other approach, we leave the notes separated.

**Pitch Sampling**  For note pitches, we use a chord-based sampling approach to introduce harmonic structure while maintaining randomness. Instead of selecting pitches arbitrarily, we begin with the tonic chord in the first bar to establish a clear tonal center. From the second bar onward, we incorporate diatonic chords, which are built solely from notes within the key's scale and do not introduce accidentals. Pitches are drawn from the selected chord, including all octave instances within the clef's pitch range (e.g., for a C note, we include C2, C3, and C4 within the allowed register). This approach ensures harmonic coherence while allowing variation, effectively mimicking real-world musical patterns and preserving both diversity and structural validity.

### 3.4 LAYOUT ADJUSTMENTS

To generate structurally diverse and visually varied music sheets, we apply a few adjustments and formatting strategies to control bar count, repetition, labeling, spacing, and note sizing. These steps ensure that the dataset captures a wide range of notation styles while adhering to common engraving practices. To maintain a single-page format, we first sample the number of bars uniformly between 10 and 20. Since various musical elements would affect the layout, we iteratively compile the latex code, adjusting the bar count as needed to ensure the output fits within a single page. Additionally, to introduce repeating sections, we randomly select two bar indices, using the smaller index as the repeat start and the larger index as the repeat end. The repeat start and repeat end symbols are then placed at the corresponding bar boundaries, reinforcing common notation patterns while enhancing structural variety. To increase presentation diversity, we independently annotate bars with chord names and/or bar indices, each with a 50% probability, allowing for cases where both, either, or neither are labeled. Furthermore, to control note compactness and visual density, we randomly select from four spacing

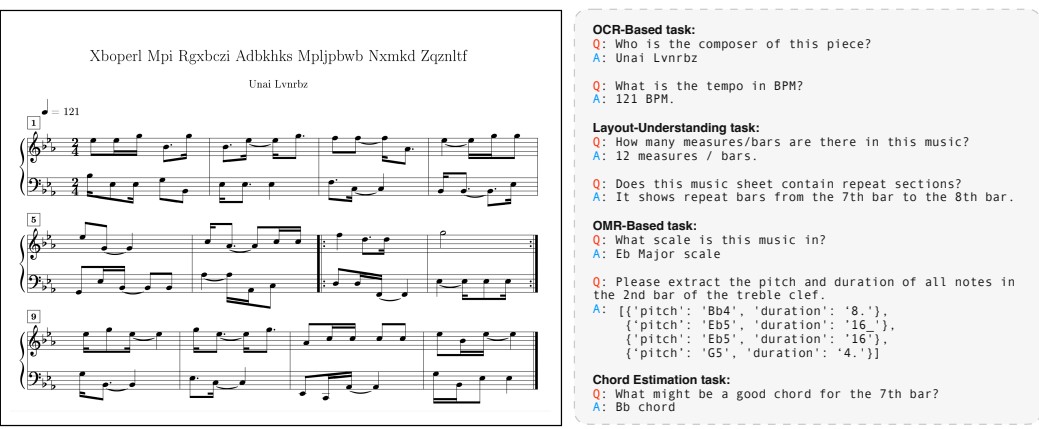

Figure 4: Example of Visual Question Answering (VQA) tasks for music sheet understanding, covering OCR and OMR-based information extraction, layout understanding, and chord estimation.

settings and two note size settings. This variation ensures that the dataset captures a broad range of engraving styles, making it more robust for training music recognition models.

## 3.5 TASK DEFINITION

We define a set of music sheet understanding tasks in a Visual Question Answering (VQA) format to evaluate Multimodal Large Language Models (MLLMs). These tasks span OCR-based text extraction, layout understanding, Optical Music Recognition (OMR), and chord estimation, requiring models to integrate visual processing with musical reasoning. To ensure precise supervision, we extract ground truth directly from MusiXTEX, avoiding post-processing errors and maintaining exact alignment with the rendered notation. The annotations are structured as question-answer (QA) pairs using task-specific templates. Figure 4 shows representative examples.

**OCR-Based Tasks** These tasks assess models' ability to extract textual information from music sheets, focusing on fundamental Optical Character Recognition (OCR) capabilities. The models are required to: (1) Identify and extract the title and author's name from the sheet. (2) Recognize the tempo marking, expressed in beats per minute (BPM). (3) Determine the time signature of the music. (4) Extract chord names when explicitly labeled in the sheet.

**OMR-Based Tasks** These tasks focus on music symbol interpretation in a specified bar and clef. The models must: (1) Recognize the key signature of the passage. (2) Extract note durations, including quarter, eighth, dotted, and tied notes, from a given bar. (3) Recognize the pitch values of individual notes. (4) Extract both duration and pitch for all notes within a specific bar and clef or across multiple clefs.

**Layout Understanding Tasks** These tasks evaluate models' ability to comprehend the structural organization of a music sheet. The model must: (1) Determine the number and type of clefs used in the sheet. (2) Count the total number of bars in the music. (3) Detect repeating sections by identifying notation patterns that indicate thematic repetition.

**Chord Estimation Tasks** For music sheets without explicitly labeled chord names, the model must infer the underlying chord based on the notes present in a given bar. This task evaluates the model's ability to understand chord structures and harmonic relationships rather than simply performing symbolic recognition.

## 4 EXPERIMENT

### 4.1 EXPERIMENT SETTINGS

To evaluate the music sheet reading capabilities of modern Multimodal Large Language Models (MLLMs), we split our dataset into training and testing partitions, with 90% of the images allocated

| Accuracy | OCR | | OMR | | Layout | | Chord | |
|---|---|---|---|---|---|---|---|---|
| | G-Acc | PNLS | G-Acc | PNLS | G-Acc | PNLS | G-Acc | PNLS |
| Paligemma2 | 4.8 | 25.8 | - | - | - | - | - | - |
| Phi-3-V | 29.8 | 54.5 | 0.4 | 7.6 | 19.2 | 49.3 | 1.7 | 74.5 |
| GPT-4o | 68.9 | 85.3 | 4.0 | 42.0 | 61.1 | 50.6 | 5.5 | 74.6 |
| GPT-4o + RAG | 69.5 | 86.9 | 3.8 | 70.7 | 61.8 | 70.4 | 5.5 | 74.6 |
| GPT-4o + RAG + OMR | 69.7 | 83.3 | 8.4 | 48.4 | 64.8 | 71.7 | 13.0 | 74.0 |
| *Finetuned on MusiXQA* | | | | | | | | |
| Phi-3-MusiX (JSON) | **97.0** | 96.1 | 9.2 | 31.1 | **94.3** | **99.5** | 19.6 | 31.1 |
| Phi-3-MusiX (kern+) | 92.6 | **99.5** | **68.4** | **99.2** | 79.6 | 95.1 | **84.9** | **96.5** |

Table 1: Quantitative comparison of six methods on the Musi𝕏QA test split. The table includes two open-source models evaluated in a zero-shot setting, and the proprietary GPT-4o model under three inference variants: zero-shot, retrieval-augmented generation (RAG), and RAG with oracle OMR results. The two Phi-3-MusiX variants are finetuned on MusiXQA using JSON and kern+ representations.

to the training set and the remaining 10% reserved for testing. We designed experiments to assess performance of 6 methods. Specifically, we evaluate Paligemma2 (3B) (Steiner et al., 2024), Phi-3-V (Abdin et al., 2024), and GPT-4o in the zero-shot setting. In addition, we introduce two GPT-4o-based baselines: (1) GPT-4o + RAG, which uses retrieval-augmented generation by retrieving the most relevant training examples to provide as in-context learning prompts, and (2) GPT-4o + RAG + OMR, which further incorporates the output of an OMR model into the prompt to enhance symbolic understanding, an effective method in OCR-related tasks (Zhang et al., 2024a;b). For retrieval, we use the image encoder from SMT (Ríos-Vila et al., 2024) to compute image embeddings and construct a similarity-based retriever. For OMR, we adopt the Oemer model (Yoyo et al., 2023) to convert music sheet images into musicxml files and extract relevant information in text format. We also propose Phi-3-MusiX, a fine-tuned version of Phi-3-V with LoRA adapters (Hu et al., 2021) trained on our dataset, highlighting the effectiveness of the dataset in enabling MLLMs to perform visual music sheet understanding. Reported results use a single prompt and training run, without average over multiple trials.

## 4.2 NOTE REPRESENTATION

We explore two text formats for representing music notes when fine-tuning our Phi-3-MusiX model for Optical Music Recognition (OMR) tasks. Both formats encode the pitch and duration of each note, but differ significantly in compactness and natural language alignment.

The **first** format is a symbolic representation we propose, called kern+, based on the **kern notation (Huron, 2002)[1]. kern+ extends the original **kern format by encoding pitches as note names with octave indices (e.g., C4) to support a wider pitch range, while preserving **kern-style duration symbols. This format is compact and well-suited for efficient token usage during training and inference. The **second** format is a JSON string, where each note is represented as a dictionary with "pitch" and "duration" keys. We use note names with octave indices (e.g., "C4") for pitch, standard note values (e.g., "8" for an eighth note) for duration, an underscore ("_") to mark the start of a slur, and a dot (".") to indicate dotted rhythms. An example is shown in Figure 4. While more verbose, this format aligns well with LLM pretraining data and supports interpretability and zero-shot generalization through its familiar structure. For evaluating baseline models, we use the JSON representation, as its code format is more aligned with the pretraining data of most LLMs and thus easier for them to interpret.

## 4.3 EVALUATION METRICS

We use Partial Normalized Levenshtein Similarity (PNLS) (Chen et al., 2024c) to evaluate model answers. PNLS computes approximate string matching with a normalized score between 0 and 1, using a partial alignment algorithm that avoids penalizing unmatched prefixes or suffixes. This makes it well-suited for comparing verbose LLM outputs against concise ground truths. Additionally, we adopt GPT-based evaluation to assess semantic correctness, recognizing that not all characters hold

---

[1]**kern representation: https://www.humdrum.org/rep/kern/

equal importance—for instance, "C major chord" vs. "D major chord" differ by one letter but convey different meanings. To address this, we prompt GPT-4o to act as a binary evaluator, assigning a score of 1 for semantically correct answers and 0 otherwise. The average score across samples is reported as GPT accuracy (G-Acc), providing a more robust measure of answer quality.

### 4.4 MODEL TRAINING

Our proposed model, Phi-3-MusiX, is fine-tuned from Phi-3-V using parameter-efficient adaptation with LoRA on the training split of the MusiⅩQA dataset. We fine-tune both the vision encoder and the language model to better align multimodal representations with symbolic music tasks. The model is trained using the HuggingFace Trainer class with mixed-precision training enabled (bfloat16). Our primary experiments are conducted on a cluster of 8 NVIDIA A100 80GB GPUs; however, the training process can also be reproduced using 48GB GPUs with appropriate gradient accumulation settings. The model is trained for 1 epoch using the AdamW optimizer (Loshchilov & Hutter, 2017), with a per-device batch size of 1 and gradient accumulation over 2 steps, resulting in an effective batch size of 16. We apply a warm-up during the first two steps and set the learning rate to $2 \times 10^{-5}$, with a weight decay of $1 \times 10^{-6}$. To ensure training stability, we apply gradient clipping with a maximum norm of 1.0.

### 4.5 MAIN RESULTS

We evaluate six methods on the test split of the MusiⅩQA dataset. Table 1 presents the results across four task types: OCR, OMR, layout understanding, and chord estimation. Performance is reported in G-Acc and PNLS.

**Open-Source Models**   Despite being trained on a large-scale dataset that includes music scores in \*\*kern representation, Paligemma2 consistently refused to answer chord estimation questions and output meaningless responses on OMR and layout tasks. This suggests insufficient adaptation to symbolic music tasks, possibly due to inadequate alignment between the training data and the QA format. Furthermore, its poor performance on the OCR task could likely be attributed to its low resolution (448x448) image encoder, which limits its ability to capture fine-grained text details.

In contrast, Phi-3-V demonstrates significantly better responsiveness and overall performance. Its relatively high scores on the OCR task are likely due to its image encoding mechanism, which splits high-resolution images into 336×336 crops, encodes each using its image encoder, and concatenates the crop features to represent the full image, allowing it to see small symbols and text more clearly.

**GPT-4o Baselines**   GPT-4o shows strong OCR capabilities, achieving high G-Acc and PNLS scores. When combined with retrieval-augmented generation (Lewis et al., 2020), the model exhibits notable improvements in PNLS for both OMR and Layout tasks. This can be attributed to in-context learning, where GPT-4o learns the expected answer format from the retrieved examples. Since the string-matching based PNLS metric is highly sensitive to formatting, even random answers with correct structural patterns can result in high PNLS scores. This behavior is further illustrated in the chord estimation task. Although models frequently predict incorrect root notes, they often append correct suffixes such as "major chord" or "minor chord". This results in a low G-Acc but a consistently high PNLS (∼0.74). Importantly, while RAG improves PNLS, it does not significantly boost G-Acc, indicating that GPT-4o fails to truly recognize music symbols but merely mimics the answer format. The baseline of 'GPT-4o + RAG + OMR', with additional context from the OMR model, slightly improves G-Acc on OMR and Chord tasks but leads to a modest decrease in PNLS. This suggests that while the symbolic information from OMR can help with correctness, the OMR input may also act as a distraction due to its length and complexity. Moreover, the Oemer model itself has limited accuracy and outputs MusicXML representations that omit key information such as accidentals (e.g., sharps and flats), requiring GPT to infer these from extracted key signatures. This adds an extra layer of reasoning, making the task more challenging and limiting performance gains from OMR input alone.

**Supervised Fine-tuning**   We fine-tuned two Phi-3-MusiX variants using different text representations for music notes: the verbose JSON format and the compact symbolic kern+ format. As shown in Table 1, the kern+ model significantly outperforms the JSON-based model on OMR and Chord tasks that require precise note-level recognition. This underscores the critical role of representation format

in structured prediction using LLMs, as observed in prior work on spatial planning with LLMs (Feng et al., 2023; Chen et al., 2024b).

To explain the observed performance difference, we interpret the output representations by categorizing text tokens into two types: format tokens, which define output format (e.g., braces, colons, and key names in JSON), and content tokens, which encode pitch and duration values. The main difference between JSON and kern+ is the ratio between this two types of tokens. For example, the note "C4" with a quarter duration is represented in JSON as `"{"pitch":"C4","duration":"4"}"`, where only "C4" and "4" are content tokens, and the remaining tokens serve formatting purposes. In contrast, the same note in kern+ is written as "qC4", using only content tokens. We believe that the dominance of format tokens in the JSON representation causes the model to converge prematurely to a local minimum, where it learns to reproduce structural patterns without improving its recognition of music notes. This often results in well-formatted but musically incorrect outputs, especially in tasks like omr and chord tasks, where accuracy depends on just a few content tokens. In contrast, the compact and content-centric kern+ format reduces structural redundancy, forcing the model to focus on informative tokens that are relevant to recognition accuracy, rather than being distracted by format tokens.

The training curves in Figure B.1 also reflect this difference. The JSON model converges quickly with stable gradients. In contrast, the kern+ model shows larger and more dynamic gradient norms, suggesting more focused learning on musical content. Finally, the Phi-3-MusiX model trained with kern+ achieves 8× and 6× improvements over the strongest GPT-4o baseline in G-Acc on OMR and Chord tasks, respectively. These results demonstrate the effectiveness of supervised adaptation and the value of MusiXQA as a training resource for symbolic music understanding in MLLMs.

### 4.6 EFFICIENCY ANALYSIS

Table 2 shows the processing time of Paligemma2, Phi-3-MusiX, and Oemer. We report per-question

| Methods | Paligemma2 | | Phi-3-MusiX | | Oemer |
|---|---|---|---|---|---|
| | OCR | OMR | OCR | OMR | |
| Time (s) | 0.45 | 1.12 | 1.16 | 1.54 | 62.34 |

Table 2: Time cost of MLLMs and the OMR Model.

time for MLLMs, and per-image time for the OMR model. In our dataset, each OMR question covers one bar and most pages have less than 20 bars, as shown in Figure 2. An entire page can be processed by iteratively querying each bar, taking about 30 seconds in the worst case and around 20 seconds on average. In contrast, Oemer takes more than a minute on average per page due to its multi-stage pipeline. This highlights the efficiency advantage of MLLMs for visual music understanding. Similar findings have been reported in previous works on document and OCR-based tasks, where end-to-end MLLMs consistently outperform pipeline-based approaches in both latency and performance (Hu et al., 2024; Liu et al., 2024b).

## 5 CONCLUSION

We introduced MusiXQA, a large-scale synthetic dataset created by rendering typeset music sheets with MusiXTEX and constructing question-answer pairs to evaluate MLLMs on symbolic music understanding in natural language. Our goal is to establish a benchmark for this emerging task, focusing on typeset music, which dominates real-world usage in digital archives and software. Experimental results show that existing models—including state-of-the-art GPT-based MLLMs—struggle with this task, highlighting its unique challenges. To demonstrate the potential of MLLMs in this domain and the value of MusiXQA, we proposed Phi-3-MusiX, a model fine-tuned on our data that achieves substantial improvements over strong baselines. Our findings further emphasize the importance of output format design in structured prediction: compact, content-focused representations like kern+ enable more effective learning than verbose formats such as JSON.

## 6 LANGUAGE MODEL USAGE STATEMENT

In preparing this manuscript, we used GPT-5 only for grammar checking and minor language polishing. The authors reviewed and edited all suggestions. All scientific content, analysis, and conclusions are entirely the work of the authors.

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

## A  SCALE DETAILS

| ♯/♭ | Root | 2nd | 3rd | 4th | 5th | 6th | 7th |
|---|---|---|---|---|---|---|---|
| *Major Scales* | | | | | | | |
| - | C | D | E | F | G | A | B |
| 1♯ | G | A | B | C | D | E | F♯ |
| 2♯ | D | E | F♯ | G | A | B | C♯ |
| 3♯ | A | B | C♯ | D | E | F♯ | G♯ |
| 4♯ | E | F♯ | G♯ | A | B | C♯ | D♯ |
| 5♯ | B | C♯ | D♯ | E | F♯ | G♯ | A♯ |
| 6♯ | F♯ | G♯ | A♯ | B | C♯ | D♯ | E♯ |
| 7♯ | C♯ | D♯ | E♯ | F♯ | G♯ | A♯ | B♯ |
| 1♭ | F | G | A | B♭ | C | D | E |
| 2♭ | B♭ | C | D | E♭ | F | G | A |
| 3♭ | E♭ | F | G | A♭ | B♭ | C | D |
| 4♭ | A♭ | B♭ | C | D♭ | E♭ | F | G |
| 5♭ | D♭ | E♭ | F | G♭ | A♭ | B♭ | C |
| 6♭ | G♭ | A♭ | B♭ | C♭ | D♭ | E♭ | F |
| 7♭ | C♭ | D♭ | E♭ | F♭ | G♭ | A♭ | Bb |
| *Minor Scales* | | | | | | | |
| - | A | B | C | D | E | F | G |
| 1♯ | E | F♯ | G | A | B | C | D |
| 2♯ | B | C♯ | D | E | F♯ | G | A |
| 3♯ | F♯ | G♯ | A | B | C♯ | D | E |
| 4♯ | C♯ | D♯ | E | F♯ | G♯ | A | B |
| 5♯ | G♯ | A♯ | B | C♯ | D♯ | E | F♯ |
| 6♯ | D♯ | E♯ | F♯ | G♯ | A♯ | B | C♯ |
| 7♯ | A♯ | B♯ | C♯ | D♯ | E♯ | F♯ | G♯ |
| 1♭ | D | E | F | G | A | B♭ | C |
| 2♭ | G | A | B♭ | C | D | E♭ | F |
| 3♭ | C | D | E♭ | F | G | A♭ | Bb |
| 4♭ | F | G | A♭ | B♭ | C | D♭ | Eb |
| 5♭ | B♭ | C | D♭ | E♭ | F | G♭ | Ab |
| 6♭ | E♭ | F | G♭ | A♭ | B♭ | C♭ | Db |
| 7♭ | A♭ | B♭ | C♭ | D♭ | E♭ | F♭ | Gb |

Table A.1: Accidentals and note composition of Major and Minor scales

## B  LOSS CURVE PLOT

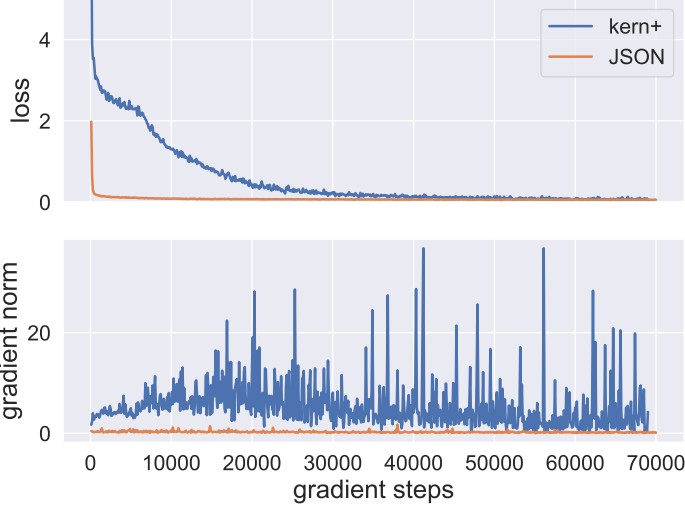

Figure B.1: Training loss and gradient norm curves for models trained with kern+ and JSON formats.

# C  SYSTEM PROMPTS

Below is the system prompt used for GPT-4o in our experiment:

*You are an AI assistant specializing in Optical Music Recognition (OMR) and Optical Character Recognition (OCR) for music sheets. Your task is to accurately analyze images of music notation and provide structured responses to visual question-answering (VQA) tasks.*

*You will process printed music sheet images and answer both OCR and OMR-related questions with high accuracy.*

*1 OCR-Based Tasks (Text Extraction)*

*- Extract the title and composer from the music sheet.*

*- Identify and extract the tempo marking (in BPM).*

*- Recognize and return the time signature.*

*- Extract explicitly labeled chord names from the sheet.*

*2 OMR-Based Tasks (Music Symbol Recognition)*

*- Identify the number and type of clefs (e.g., treble, bass).*

*- Count the number of bars (measures) in the music sheet.*

*- Recognize repeat sections based on notation symbols.*

*- Extract note durations (e.g., quarter, eighth, dotted notes, tied notes) for a given bar.*

*- Identify note pitches within a given bar.*

*- Return a structured representation of pitch, duration for a given bar in JSON string of list of python dictionaries without indent.*

*- Use kern representation for duration.*

*- If no explicit chord labels exist, infer the chord based on the notes in a given bar.*

*3 Response Format*

*- Provide structured, precise, and as concise as possible answers.*

*- Use structured JSON output without indent, when applicable for easy parsing.*

*4 Additional Considerations*

*- Ensure responses are notation-aware, considering key signatures, accidentals, and note relationships.*

*- Handle staff line separation correctly, ensuring multi-clef scores are properly analyzed.*

*- Avoid hallucinating missing information; only extract what is present in the image.*

*Follow music engraving conventions and OMR best practices to provide accurate, structured answers. If the requested information is not visible in the image, respond with '"Information not found"' instead of making assumptions.*

