# OpenReview forum: "MusiXQA: Advancing Visual Music Understanding in Multimodal LLMs"
_ICLR.cc/2026/Conference — Submitted to ICLR 2026_

### Official Review · Reviewer_u5jf · 2025-10-29

**Soundness:** 3
**Presentation:** 3
**Contribution:** 3
**Rating:** 8
**Confidence:** 4

**Summary:**

This paper tackles the inability of current MLLMs to understand sheet music by introducing MusiXQA, the first comprehensive visual question-answering dataset for this task. It features high-quality synthetic music sheets generated via MusiXTEX, paired with diverse QA tasks covering pitch, chords, and more. Based on this data, the authors' fine-tuned Phi-3-MusiX model significantly outperforms baselines like GPT, demonstrating the dataset's effectiveness in advancing music sheet understanding.

**Strengths:**

1. The experimental design is excellent. By using two distinct metrics—one for format (PNLS) and one for semantic accuracy (G-Acc)—the authors cleverly demonstrate that powerful baselines like GPT-4o were not truly understanding the music but merely "mimicking" the format of the correct answer .
2. The paper is the first to systematically address a clear and challenging gap in AI: the inability of advanced MLLMs to read and understand symbolic sheet music. It introduces a comprehensive benchmark, MusiXQA, to drive progress in this new area.

**Weaknesses:**

1. The model's performance on noisy, real-world scanned music is unknown, as it was trained exclusively on "perfect" synthetic data, creating a significant "sim-to-real" gap.
2. The OMR-assisted baseline (GPT-4o + RAG + OMR) is weak because it relies on an OMR tool that the authors admit has "limited accuracy" and omits critical information like accidentals.
3. The dataset's musical diversity is limited, as it intentionally excludes common elements like $6/8$ time 3and relies on randomly generated, musically incoherent notes rather than real compositions 4.

**Questions:**

see Weaknesses.

---

> ### Author Response · Authors · 2025-11-25
>
> Thank you very much for your thoughtful and encouraging feedback. We sincerely appreciate your recognition of the problem’s importance and the strengths of our experimental design. Below we address your concerns.
>
> **W1**. Concern about sim-to-real gap due to training only on clean synthetic data
>
> **A1**. We agree this is an important concern. As described in our **global response A3**, we conducted an additional experiment by printing 20 randomly selected synthetic sheets and photographing them to introduce real-world distortions. Despite being trained exclusively on clean images, Phi-3-MusiX retained partial robustness inherited from the pretrained vision encoder. We believe this result strengthens the paper, and we will include the new findings and discussion in the revised version.
>
>
> **W2**. OMR-assisted baseline is weak due to limited OMR performance
>
> **A2**. Thank you for pointing this out. We selected Oemer because it is a recent (2023), easy-to-use OMR system and, among the tools we tested, the only one that produced outputs consistently on our dataset. Many existing OMR models are not designed for full-page inputs, and their training corpora lack the key-signature coverage of MusiXQA; as a result, several systems produced incorrect outputs or raised errors. These limitations further highlight the value of MusiXQA. If we identify an OMR system that can process our dataset reliably, we will include it in our final version. Regardless, the latency and memory-cost observations in our paper remain valid, since they stem from structural characteristics of OMR-assisted pipelines.
>
>
> **W3**. Limited musical diversity.
>
> **A3**. Thank you for pointing this out. The current symbolic scope is limited primarily by MusiXTEX, the open-source and scriptable typesetting system we use. MusiXTEX’s engraving controls are limited and sparsely documented, which restricts the range of notation we can reliably generate. The framework itself is renderer-agnostic, and the backend can be replaced with more expressive notation engines in future work to naturally expand symbolic and musical diversity.

---

### Official Review · Reviewer_NL27 · 2025-10-30

**Soundness:** 2
**Presentation:** 3
**Contribution:** 2
**Rating:** 2
**Confidence:** 4

**Summary:**

This paper investigates the effectiveness of applying VLMs to music sheet understanding. Authors constructed a synthetic music sheet
dataset MusiXQA containing structured annotations. VQA-like tasks are used to train and evaluation VLMs' ability on understanding music
sheets. Specifically, the tasks of the annotations include information extraction, sheet layout understanding, and chord estimation etc.

Authors show that SOTA VLMs perform badly on the MusicXQA benchmark, and then finetuned a Phi3 model on MusicXQA, achieving the best performance.

**Strengths:**

- The paper is well structured, and easy to follow. Experiment and results are well presented.
- Creating resources and datasets for the music sheet understanding task, which is resource-lean, is appreciated.

**Weaknesses:**

- The MusicXQA dataset is completely synthetic (e.g., music title, composer name etc.), and I am concerned about real-world performances of a model trained and evaluated on MusiXQA. For example, authors mentioned that their created dataset "do not need to be musically coherent or aesthetically refined", as a result, I think this makes MusicXQA tasks more like general segmentation or extraction tasks for symbolic music notes, orthogonal to music understanding. So I am concerned about thow MusicXQA helps creating strong music understanding models.

- When randomly generating MusicXQA, some settings/parameters seem used ad hoc, without supporting evidences (cf questions).

- Authors finetuned a Phi3 model using MusicXQA training data and show it performs the best on MusicXQA test data. This is very much   expected and unfair to other baselines. I think it is needed to also evaluate the models in OOD settings, such as MMMU's music sheet questions.

**Questions:**

- Line207: how did you decide to use 1-10 words for the title and 1-3 words for author names? Do these reflect the real-world cases?

- Line232: how did you decide to use the number of bars (10-20) and number of notes?

- Line269: why using four spacing settings and two note size settings?

---

> ### Author Response · Authors · 2025-11-25
>
> Thank you very much for your thoughtful feedback. We apologize if our explanations were not sufficiently clear and may have caused some misunderstandings, and we hope the clarifications below will address your concerns.
>
> **W1.1**. Difference between our task and symbolic music notes segmentation or extraction.
>
> **A1.1**. Thank you for raising this concern, and we are sorry for any confusion. We would like to clarify the distinction between our goal and general symbolic music-note segmentation or extraction tasks. As stated in our **global response A1**, our work investigates whether MLLMs can support **conversational interaction** over music sheets and answer questions in the **human readable text**, rather than performing deterministic segmentation or extraction like traditional OCR/OMR models that output structured metadata (bounding boxes, labels, confidence scores) for machine consumption, requiring compatible downstream code or pipelines.
>
> **W1.2**. Whether our task is orthogonal to music understanding and how MusicXQA helps create strong music understanding models.
>
> **A1.2**. we believe it is an essential component required for MLLMs to perform **visual music understanding in conversation**. We decompose this capability into three components: (i) visual perception, (ii) visual–text alignment in the hidden representation space, and (iii) music understanding/reasoning in linguistic space. This decomposition is inspired by the observation we discussed in **global response A1**: models such as GPT already show strong music-theory understanding in the text modality, yet they typically refuse to answer when the same information is presented visually as music sheets images. This observation indicates that the missing part is  (i) **visual perception** and (ii) **visual-text alignment**, not musical knowledge. MusiXQA therefore focuses on these two parts, enabling MLLMs to see music notation and connect visual symbols to their pretrained language token space. As explained in **global response A2**, this requires large-scale, clean, detailed VQA-style annotations, which motivates the creation of synthetic data.
>
>
> **W1.3**. Concerns about real-world generalization.
>
> **A1.3**. We appreciate the concern about real-world performance. As detailed in our **global response A3**, we conducted an additional experiment by printing randomly selected synthetic sheets and photographing them to introduce real-world distortions. Despite being trained only on clean synthetic images, Phi-3-MusiX retained partial robustness inherited from the pretrained weights, indicating that the visual-perception capabilities learned from synthetic data do generalize to some extent to real-world photographic variations. However, due to the limited rendering capabilities of MusiXTEX, not all music symbols/layout are included in our dataset, thus, the model trained only on MusiXQA won’t be able to process unseen content.
>
> **W2/Q1,2,3**. Reason for the parameter choices in our synthetic data generation process.
>
> **A2**. The choices for the title and author-name lengths are indeed simple, experience-driven decisions. These fields are included primarily to preserve the model’s existing text-reading (OCR) ability. We only adopted minimal assumptions—such as titles tending to be longer and placed above the author name, and author names typically containing no more than three words. Most of the other simplified settings are due to the limitations of the typesetting system MusiXTEX. For example, MusiXTEX provides only four spacing presets and two note-size settings, which makes it infeasible to render more than 20 bars on a single page without causing layout instability. Since our focus is on single-page image understanding, we choose 10–20 bars to satisfy single-page constraints while avoiding large blank regions that provide limited training value.

---

> ### Author Response · Authors · 2025-11-25
>
> **W3**. Is the good performance on clean synthetic data expected? And can model performance well on MMMU?
>
> **A3**. The good performance on clean synthetic data is not guaranteed or trivial. As shown in **Section 4.5 (Table 1)**, even with large-scale clean supervision, exhaustive combinatorial symbolic coverage, and days of training, the model still fails to learn OMR/Chord tasks when the music text representation is inappropriate. This demonstrates that the learning problem is fundamentally **non-trivial** and highly sensitive to representation design.
>
> Since this work is the first to introduce and study this task, our fine-tuned Phi-3-MusiX serves as a **feasibility validation** of the **dataset design** and **methodological choices**, rather than a model intended for real-world generalization. In addition, there is currently no suitable baseline: even GPT-4o performs poorly when answering these VQA-style music-sheet questions. In this context, fine-tuning on a dataset and evaluating on the same distribution is the standard procedure for verifying whether the dataset provides the necessary learning signals.
>
> Regarding MMMU, its music-sheet questions come from a very different and highly sparse distribution, and it remains an unsolved and extremely challenging test set. To the best of our knowledge, there is no available training dataset that enables MLLMs to handle this subset robustly. Therefore, performance on MMMU would not meaningfully reflect the effectiveness of our visual-symbol learning framework.

---

> > ### Comment · Reviewer_NL27 · 2025-11-27
> >
> > Thank you for your response, discussions, and sharing details
> > about the parameter choices during the data generation process.
> >
> > However, I am still concerned about the synthetic nature of MusiXQA
> > and its real-world applications. As a result, I would like to keep my
> > current rating.

---

> > > ### Author Response · Authors · 2025-11-28
> > >
> > > Thank you for the follow-up and for considering our responses. We appreciate your time and effort in reviewing our work.

---

### Official Review · Reviewer_ZSTE · 2025-11-01

**Soundness:** 3
**Presentation:** 3
**Contribution:** 2
**Rating:** 6
**Confidence:** 4

**Summary:**

This work aims dealing with a specific multi-modal task, which is interpret and understand music sheet images. A dataset, MusiXQA is proposed to help training MLLMs on this task. In the end, Phi-3-MusiX, a fine-tuned version of Phi-3-Vision, is proposed and demonstrated its performance advantage across four types of music sheet understanding tasks.

**Strengths:**

- Well-motivated problem setup in a specific domain that is not resolved yet but demanding for people working on music analysis.

- No real-world MIDI used, therefore no issues on publishing rights.

- The task definition in section 3.5 is well categorized and covered the necessary components of the entire workflow of "music sheet interpretation". Especially the OMR-based tasks and Chord estimations tasks. The former checks symbolic understanding and latter can evaluate semantic understanding.

- According to Table 1, fine-tuning Phi-3-Vision with MusiXQA indeed yield huge improvement on all the four types of tasks. Demonstrating the usefulness of the proposed dataset.

- The subsection about performance difference between JSON/kern+ is insightful. On the other hand, if semantic density is the key, it's interesting know if any alternative can serve the purpose.

**Weaknesses:**

- Since all the MusiXQA images are all generated by MusiXTEX, a concern is that the image encoder trained on this dataset may lack generalizability for sheet images in general. For example, size of notes, fonts, minor style variation, or hand-writing sheets.

- Some design of MusiXQA seems to be too restrictive and can only cover a limited subset of music sheets. For example, the pitch range of A\flat1 ~ F\sharp6 cannot cover the note range of tuba, bass, violin or piccolo. More examples are:
a) Each word in the name is limited to 3-8 characters.
b) Seems ruling out orchestral music by assuming only two staves.
c) The BPM range of 50-140 could not cover certain music styles and instrument sheets.
d) Time signature is limited to 2/4, 3/4, 4/4.
e) Seems do not support scale/tone switching
f) Only support up to 1/16th notes.
g) Always begin with the tonic chord of the key makes the generation easier, but this is not the case in real-world and could cause over-fitting for models trained on such data.
h) no accidentals

It's understandable that some of the decision is limited by the need to obtain the semantic ground truth from randomly generated score. Would it be better if creating the dataset in two parts, the first part includes as much as variations for symbolic understanding, and the second part is a subset of the first and accompanied with QA pairs for semantic understanding tasks?

**Questions:**

- In section 4.1, about RAG-like in-context learning, how does the "most relevant" training example is determined?

- In section 4.3, is there any procedure to validate that the evaluation of GTP-4o is correct at most of the time?

- Table 1: Is there any insight on why the model trained on kern+ has much worse G-Acc on Layout understanding task?

---

> ### Author Response · Authors · 2025-11-24
>
> We thank the reviewer for the thoughtful and detailed feedback. We appreciate the constructive suggestions and insightful questions, and we address them point-by-point below.
>
> **W1**. Generalizability concern due to fully synthetic MusiXTEX images.
>
> **A1**. We are sorry for any confusion regarding the scope of our task and goal. As clarified in our **global response A1**, our work focuses on whether MLLMs can acquire **visual symbolic perception** and **cross-modal alignment** under **clean, precisely aligned supervision** to support conversational interaction, rather than on building a production-level assistant. We also show that this learning problem is non-trivial: even with large-scale clean data, the model fails when the output representation is inappropriate. Thus, our contribution centers on understanding how visual perception and cross-modality alignment can be learned, rather than on broad real-world generalization (e.g., handwritten or stylistically diverse scores). We provide more discussion in global response A1 and warmly invite you to take a look. Although generalization is not our central objective, we still conducted an additional experiment motivated by these concerns.
>
> As summarized in our **global response A3**, we consider two types of generalization. For **photographic distortions**, our printed-and-photographed experiment shows partial robustness inherited from the pretrained Phi-3-V encoder. For **out-of-distribution symbolic variety**, the model cannot generalize to notation elements never appearing in training—this reflects limitations of MusiXTEX rather than of our framework. Because our pipeline is modular, MusiXTEX can be replaced in future work by more capable typesetting systems (via APIs or GUI-agent automation) to expand symbolic coverage without changing the methodology.
>
> **W2**. Several design constraints in MusiXQA may restrict the dataset to a limited subset of real-world music sheets.
>
> **A2**. We appreciate your detailed observations regarding the design constraints. As noted in our **global response A1**, these constraints do not impact our scientific goal, since the task primarily focuses on enabling **visual perception** and **visual–text alignment**, rather than injecting additional musical knowledge or reasoning ability. The distinction between these two aspects is discussed further in **global response A1**.
>
> Most of the simplified configurations arise from practical limitations of the typesetting system MusiXTEX, which provides limited programmatic control over complex engraving. To ensure clean rendering for **reliable ground-truth extraction**, we avoid settings that frequently lead to low-quality layouts. For example, extreme pitches often cause overlapping ledger lines or unstable spacing. Within these constraints, the adopted pitch range (A♭1–F♯6) still covers widely used modern instruments such as electric guitar (E2–E6) and electric bass (A1–A4), which are highly representative in small band settings and often carry key melodic and rhythmic roles.
> Importantly, these limitations reflect MusiXTEX rather than our framework. The pipeline is fully modular, and more diverse configurations can be supported by simply swapping in a more capable typesetting engine in future work, without altering the core methodology, as discussed in the second paragraph of our **global response A3**.
>
> **Reviewer Suggestion**: Consider creating the dataset in two parts: a broad symbolic-variation set and a subset with semantic QA.
>
> **A**. Thank you for this insightful suggestion. We agree that expanding MusiXQA with a second subset targeting deeper, music-theory-oriented semantic QA is indeed valuable. Creating such QA pairs, however, requires substantial domain expertise, since higher-level questions (e.g., harmonic implications, motivic relations, phrase-level structures) require musically meaningful content in the score and cannot be reliably annotated through simple programmatic rules.
>
> Nevertheless, motivated by your suggestion, we conducted a small exploratory attempt by providing ground-truth melodies in text form to GPT-5.1 and prompting it to generate more advanced music-theory QA. The generated QA varied in correctness and depth, and our model showed limited effectiveness on these questions, likely because the LLM backbone, Phi-3, acquired limited music-theory knowledge during pretraining. This suggests that supporting deeper semantic reasoning may require additional finetuning on text-domain music-theory QA datasets, rather than modifications to the visual-perception pipeline.
>
> These findings reinforce your intuition: a deeper semantic QA subset is feasible and promising, but its construction requires careful annotation design and likely LLM-side adaptation. We view this as a meaningful and exciting direction for future extensions of MusiXQA.

---

> ### Author Response · Authors · 2025-11-24
>
> **Q1**. How is “most relevant example” determined in Section 4.1?
>
> **A**: Thank you for the question. As stated in lines 346–348 of the submission, we compute image embeddings using the SMT image encoder and retrieve the nearest training example based on cosine similarity. We use SMT because it is a recent vision encoder trained directly on music sheet images, making it more suitable for computing relevance in this domain. We will revise the text to make this procedure clearer and easier to follow.
>
> **Q2**. In section 4.3, is there any procedure to validate that the evaluation of GTP-4o is correct at most of the time?
>
> **A**. In our evaluation, GPT-4o is used only as a semantic-equivalence judge, which is far simpler than solving the underlying music-understanding question. This practice is widely adopted in recent QA and VQA literature. While GPT-based evaluators are not guaranteed to be 100% correct, prior work [1] has shown that GPT’s semantic-equivalence scores strongly correlate with human judgments, making it effective for qualitative comparison between natural-language answers. We will add citations in the revision to reflect its common use as an automatic evaluation metric.
>
> [1]. Zheng, Lianmin, et al. "Judging llm-as-a-judge with mt-bench and chatbot arena." Advances in neural information processing systems 36 (2023): 46595-46623.
>
> **Q3**. Table 1: Is there any insight on why the model trained on kern+ has much worse G-Acc on Layout understanding task?
>
> **A**. Thank you for the question. The lower G-Acc of the kern+-trained model on the Layout Understanding task is primarily due to the different optimization dynamics induced by the two output representations.
>
> For the kern+ model, the representation forces the model to learn the perception of music notes and accidentals—the visually complex “tadpole-shaped” symbols—which the JSON-trained model never learns. These music-symbol tokens contribute significantly to the loss, so the model must devote most of its learning capacity to acquiring symbolic-reading ability. In our experiments, the kern+ model required over a week of training on 8×A100 GPUs before the loss decreased sufficiently and the model reliably learned to read notes. Because this symbolic-learning stage is long and computationally demanding, gradients from OMR- and chord-related QA dominate early training, and layout-related learning signals are diluted.
>
> In contrast, in JSON training, we observed a much faster loss reduction. This occurs because the model quickly overfits to the JSON format itself: the majority of tokens are structural and easy to predict, and music-symbol tokens contribute only minimally to the overall loss. Since OCR ability already exists in the pretrained weights, the model is effectively updated mainly through gradients from layout-related QA.
>
> Thus, the observed gap does not indicate that the kern+ model is inherently worse at layout understanding. It simply reflects that the two models are at different points in their learning trajectory under equal training time. With a longer training schedule, a higher proportion of layout QA, or a higher-rank LoRA, we expect the layout performance of the kern+ model to reach the level of the JSON model. The current results reflect that the kern+ model prioritized the far more challenging symbolic-reading tasks during the available training steps.

---

### Official Review · Reviewer_UrLQ · 2025-11-11

**Soundness:** 3
**Presentation:** 3
**Contribution:** 2
**Rating:** 4
**Confidence:** 3

**Summary:**

This paper addresses the underexplored capability of Multimodal Large Language Models (MLLMs) to understand music sheets. The authors introduce MusiXQA, a new, large-scale benchmark dataset for this task. The dataset contains 9,600 high-quality synthetic music sheets generated using MusiXTEX, paired with over 130,000 visual question-answering (VQA) pairs. The VQA tasks cover four categories: OCR, Layout Understanding, Optical Music Recognition (OMR), and Chord Estimation.
The authors evaluate current SOTA MLLMs (like GPT-4o and Paligemma2) and find they perform very poorly, with GPT-4o failing on OMR tasks and Paligemma2 refusing to answer them entirely. To demonstrate the value of their dataset, they fine-tune a Phi-3-Vision model, creating Phi-3-MusiX. This model achieves significant performance gains, such as an 8x improvement in G-Acc on OMR tasks over the best GPT-based baseline. A key finding is that a compact, symbolic output representation (kern+) vastly outperforms a verbose JSON format , as the latter causes the model to learn "format tokens" rather than "content tokens".

**Strengths:**

- Identifies a clear and difficult gap in the capabilities of current SOTA MLLMs.
- Introduces MusiXQA, the first large-scale VQA benchmark specifically designed for music sheet understanding in MLLMs.
- Provides an excellent and insightful analysis of output representation, demonstrating that compact, symbolic formats (kern+) are far superior to verbose formats (JSON) for fine-tuning MLLMs on structured prediction tasks.
- The fine-tuned Phi-3-MusiX model shows massive (8x) performance gains over SOTA baselines, proving the dataset's value for domain-specific adaptation.

**Weaknesses:**

- Synthetic Data Limitation: The paper's main weakness is that the MusiXQA dataset is purely synthetic, generated from MusiXTEX. This ignores the primary challenges of real-world OMR, such as scanned artifacts, image noise, and handwritten notation, which are addressed by prior OMR datasets (e.g., CVC-MUSCIMA).
- Poor Generalization: The paper positions MLLMs as a promising alternative to OMR pipelines. However, a model (Phi-3-MusiX) trained exclusively on this pristine, synthetic data is highly unlikely to generalize to real-world scanned or handwritten scores. The paper benchmarks a "clean" problem, not the real, messy one.
- Unfair Efficiency Comparison: The paper claims an efficiency advantage over the OMR tool Oemer. This is an apples-to-oranges comparison. Oemer is designed to handle the much harder task of real-world (and likely distorted) scans, while the MLLM is only processing its own clean, synthetic training distribution.

**Questions:**

- The core weakness is the synthetic nature of the data. How do the authors expect Phi-3-MusiX, trained only on MusiXQA, to perform on real-world scanned music sheets or on handwritten OMR datasets like CVC-MUSCIMA?
- Given the lack of real-world noise, isn't the claim of MLLMs being a promising alternative to OMR pipelines  premature?
- Could the MusiXTEX generation framework be extended to simulate real-world distortions (e.g., page curl, scanner noise, faded ink, slight rotations) to help bridge this synthetic-to-real gap?

---

> ### Author Response · Authors · 2025-11-24
>
> Thank you for your insightful and constructive comments. We are sorry for any confusion caused, and we include our detailed responses below to address your questions and concerns.
>
> **W1 + Q2**. Concerns about using synthetic data and difference from “traditional OMR”.
>
> **A1**. As we explain in our **global response A1**, our goal is to investigate whether MLLMs can acquire **visual symbolic perception** and **cross-modal alignment** needed for **conversational interaction with users**, which is fundamentally different from traditional OMR pipelines, which focus on **machine-oriented symbolic extraction** for downstream processing. This distinction requires **VQA-style annotations** and a **large-scale**, **precisely aligned** dataset, which real scanned datasets cannot provide reliably or at scale. For this reason, synthetic MusiXTEX data is essential for offering **clean supervision** and **combinatorial coverage**, both of which are important for learning the capability we study. We provide additional discussion of these points in our **global response A1 and A2**, and we warmly invite you to take a look if you are interested in more details.
>
> **W2 + Q1 + Q3**. Concerns about real-world generalization, scanned distortions, and handwritten scores.
>
> **A2**. Thank you for your insightful and constructive suggestion in Q3. Following your recommendation, we conducted an additional experiment by printing 20 randomly selected synthetic music sheets and photographing them with a smartphone to introduce real-world distortions such as lighting variation, color shifts, and mild geometric or perspective changes. As shown in our **global response A3**, the model retains partial robustness—likely inherited from the pretrained Phi-3-V vision encoder—despite being fine-tuned only on clean synthetic images. We believe this experiment further strengthens the work, and we will include the new results and discussion in the final version.
>
> Beyond this experiment, we would also like to clarify that our objective is dataset design, methodological exploration, and feasibility analysis, rather than building a production-level MLLM-based music assistant. The model is therefore not expected to fully generalize to all real-world or handwritten scenarios; instead, it serves as an initial step toward understanding how symbolic visual perception can be learned under controlled supervision. As discussed in our global response A2, typeset notation is the dominant format in modern music practice, making it a natural and practical starting point before extending to handwritten or more diverse real-world styles.
>
> **W3**. Unfair Efficiency Comparison.
>
> **A3**. Thank you for raising this point. As clarified in our **global response A1**, our intention was not to claim that MLLMs are inherently more efficient than traditional OMR systems. Rather, we aim to highlight a phenomenon analogous to the transition from OCR-integrated LLMs to end-to-end MLLM pipelines in text-rich document understanding. When an external optical recognition tool is integrated into an MLLM workflow for conversational interaction, it typically processes the **entire page**, not just the content relevant to the query, producing a **long token sequence** as input of the LLM. This leads to both **high memory usage** and additional **latency**, regardless of recognition accuracy.
>
> The goal of our efficiency experiment is therefore to illustrate the **efficiency cost** of relying on an external OMR tool, rather than to compare OMR and MLLM pipelines as if they were alternative approaches. We will revise the final version to avoid any impression of an efficiency comparison between mismatched settings.

---

### Author Response · Authors · 2025-11-24
**Global Response**

We thank all reviewers for their careful reading and constructive feedback. We are encouraged that the manuscript was found to be well motivated, well structured, and clearly written, and that reviewers recognized the novelty of introducing a new task setting involving visual symbolic music understanding in MLLMs together with a dataset specifically designed to evaluate this capability. The reviewers also raised several common concerns. We address these shared issues below and then respond to reviewer-specific questions individually.

---

> ### Author Response · Authors · 2025-11-24
>
> **Q1**. What exactly is the task we study, and how it differs from traditional OMR
>
> **A1**. Our work investigates whether modern MLLM architectures can acquire visual music understanding through **conversational interaction** when provided with large-scale, clean, and precisely aligned supervision. This task is motivated by a clear empirical observation: current proprietary MLLMs (e.g., GPT-4o, Claude) show a clear gap between their strong music-theory reasoning in text and their inability to interpret symbolic notation from images or PDFs. This gap indicates that the core bottleneck is not musical reasoning itself, but the model’s ability to perform **visual symbolic perception** and **cross-modal alignment**.
>
> This task differs fundamentally from OMR. OMR systems operate like object detection or OCR models: they analyze the entire image and produce structured outputs intended for downstream machine processing rather than natural-language responses for human users, as modern LLM-based assistants would do. This mirrors the historical progression in document understanding, where early multimodal models relied on OCR outputs as auxiliary text, which introduced significant latency and memory overhead until large-scale training enabled fully **end-to-end** reading. Symbolic music perception is even more challenging than text recognition because musical symbols are dense, highly similar in appearance, and depend on subtle spatial relationships. Our study therefore investigates whether MLLMs can acquire symbolic music perception end-to-end without relying on an external OMR tool.
>
> Our results also show that direct supervised fine-tuning on clean data is not sufficient by itself, and that an appropriate text representation is crucial for effective learning.
>
> **Q2**. Why synthetic data is necessary
>
> **A2**. Because our goal is visual music understanding through conversational interaction, we require a large-scale dataset with fine-grained **VQA annotations** and **combinatorial coverage** over musical symbols and attributes. Such controlled variation is necessary for isolating concepts that are entangled in real-world corpora, which existing scanned or handwritten datasets cannot provide. In contrast, our controlled generation pipeline offers **exhaustive symbolic variation** with perfectly aligned ground truth. This limitation also appears empirically: as shown in Experiment 4.5 (Table 1), the OMR model (Oemer) performs poorly even on clean typeset images due to the limited and biased annotations in its training data. Finally, typeset notation is the dominant representation in modern music practice (Sibelius, Finale, Dorico, MuseScore, Guitar Pro) and the standard medium through which performers exchange scores. Clean typeset notation is therefore a practical and appropriate starting point for studying symbolic perception.

---

> ### Author Response · Authors · 2025-11-24
>
> **Q3**. Can the model generalize to real-world data?
>
> **A3**. Real-world difficulty arises from two sources:
>
> (1) **Visual distortions** introduced by cameras or scanners.
>
> To assess the first aspect, we conducted a small additional experiment inspired by reviewer UrLQ’s suggestion. We randomly sampled 20 MusiXTEX-generated sheets, printed them, captured them using a smartphone (we have uploaded to Supplementary Material), and evaluated the kern+-trained Phi-3-MusiX model. Although both GPT-score and string-matching accuracy decreased slightly, they remained substantially higher than all baselines, indicating that the model retains partial **robustness** to photographic distortions from Phi-3-V, even though MusiXQA itself contains only clean typeset images.
>
> Performance of Phi-3-MusiX trained using the kern+ representation:
>
> |Image |OCR||OMR||Layout||chord||
> |--|--|--|--|--|--|--|--|--|
> ||G-Acc|PNLS|G-Acc|PNLS|G-Acc|PNLS|G-Acc|PNLS|
> |Snthetic image|80.0|98.8|90.0|97.1|80.0|97.2|90.0|98.6|
> |Photo of printed image|76.2|96.4|85.0|93.8|70.0|83.3|70.0|91.3|
>
> (2) **Out-of-distribution symbolic variety**.
>
> The second aspect—symbolic variety—is constrained primarily by MusiXTEX, whose incomplete documentation and limited rendering capabilities restrict the set of notation elements that can be reliably generated. Consequently, MusiXQA does not cover all possible symbolic or layout variations, and Phi-3-MusiX should not be expected to generalize to unseen notation styles. This is a limitation of the rendering engine rather than of our framework: the pipeline is modular, and the rendering backend can be replaced with more capable typesetting systems (e.g., Dorico, Finale, Sibelius, Guitar Pro) once API access or GUI-agent automation becomes available, thereby expanding the symbolic space without altering the methodology.
>
> Finally, the goal of this work is **dataset design**, **methodological exploration**, and **feasibility analysis**, rather than proposing a production-level assistant that generalizes to all real-world scenarios. We contribute a dataset that enables systematic training of MLLMs, which is not available in existing resources, and we provide empirical insights into how symbolic text representations should be designed for effective multimodal learning.

---

### Author Response · Authors · 2025-12-03
**Global Follow-Up**

As the discussion period concludes, we sincerely thank all reviewers for their time and constructive feedback. We hope our responses have clarified the scope of the work and addressed the main concerns.

We would like to briefly restate that our submission is in the **Primary Area: Datasets and Benchmarks**. MusiXQA is designed as a foundational resource for studying visual symbolic music understanding—a capability that current MLLMs lack and for which **no aligned, VQA-style corpus exists**. Precisely annotated symbolic music images cannot be reliably produced by existing models or OMR systems at scale and with sufficient distributional coverage, making synthetic but clean, domain-informed data the only feasible path. MusiXQA fills this gap and **does not overlap with any existing dataset**.

As with many historical ML milestones, carefully designed synthetic datasets (e.g., CLEVR, SCAN, Atari, ShapeNet) have enabled new research directions by isolating core representational challenges. We believe MusiXQA serves a similar role for symbolic music perception.

Beyond creating the dataset, we fine-tuned Phi-3-MusiX to verify that MusiXQA provides meaningful learning signals, and we report **training insights** such as the importance of **compact symbolic representations**. These experiments are not intended as a complete solution, but as evidence that the dataset is **effective** and **scientifically valuable**.

We respectfully note that expectations of broad real-world generalization reflect an evaluation lens appropriate for full task-solving method papers, whereas our model—trained solely on a newly introduced dataset that fills a previously unexplored space—is not intended to serve as such a solution. Imposing this requirement therefore falls outside the expectations and scope of a dataset-track submission.

---

### Meta-Review · Area_Chair_upZM · 2026-01-07

**Summary:**

Reviewers acknowledge the novelty of introducing MusiXQA as the first VQA benchmark for music sheet understanding in MLLMs, the clear experimental setup, and valuable insights on compact output representations (kern+). However, major concerns center on the exclusive use of synthetic data, limited symbolic coverage due to MusiXTeX constraints, and insufficient demonstration of generalization to real-world conditions (noise, scans, handwritten scores, broader notation).

While the rebuttal effectively clarifies task scope and provides new experiments showing partial robustness to printed/photographed sheets, the core limitation remains: the benchmark and model are evaluated primarily on clean, synthetic typeset notation. To the AC however, demonstrating relevance to practical music sheet understanding—including common real-world variations—is critical. The analogy to CLEVR is noted, but CLEVR was rapidly solved (see https://arxiv.org/abs/1709.07871 or https://arxiv.org/abs/1706.01427) and while being popular for a while, is no longer relevant as a core diagnostic for modern MLLMs, thus reducing its persuasiveness as justification for accepting a similarly constrained synthetic benchmark.

Therefore the decision to reject, especially given the calibration among papers received. Nevertheless, the AC encourages authors to submit the strengthened version to a future conference, ideally to a dataset/benchmark track.

**Reviewer Concerns:**

Mitigated:

- Task distinction from traditional OMR.
- Necessity of synthetic data for clean, large-scale VQA supervision.
- Partial robustness to photographic distortions.
- Efficiency comparison.
- Parameter choices justified by MusiXTeX limits.

Remaining Concerns:

- Full generalization to noisy/handwritten/real-world compositions.
- Broader symbolic/musical diversity beyond MusiXTeX renderer constraints.
- Limited real-world relevance (scans, artifacts, diverse notation styles).

**Reviewer Scores:**

- Reviewer UrLQ (original 4, marginally below): Likely remain 4.
- Reviewer ZSTE (original 6, marginally above): Likely remain 6.
- Reviewer NL27 (original 2, reject): Likely unchanged; maintained low rating post-rebuttal due to persistent real-world concerns.
- Reviewer u5jf (original 8, accept poster): Likely remain 8.

---

### Decision · Program_Chairs · 2026-01-26

Reject